# Profiles of expression pattern and tissue distribution of host defense peptides genes in different chicken (*Gallus gallus*) breeds related to body weight

Zhengtian Li[1], Irfan Ahmed[1], Zhiqiang Xu[1,2], Shuai Sun[1], Tao Li[1], Dahai Gu[1,2], Yong Liu[1], Xia Zhang[1], Shixiong Yan[1], Wenyuan Hu[1], Ziqing Jiang[1], Xiaohua Duan[1], Qihua Li[1], Lixian Liu[1], Hua Rong[1], Kun Wang[1], Alsoufi Mohammed Abdulwahid[1], Tengfei Dou[1], Shanrong Wang[1], Zhengchang Su[3], Changrong Ge[1], Marinus F. W. te Pas[4,5]*, Junjing Jia[1]*, Ying Huang[1]*

1 Yunnan Provincial Key Laboratory of Animal Nutrition and Feed, Yunnan Agricultural University, Kunming, Yunnan Province, People's Republic of China, 2 Department of Food Science, Yunnan Agricultural University, Kunming, Yunnan Province, People's Republic of China, 3 Department of Bioinformatics and Genomics, College of Computing and Informatics, The University of North Carolina at Charlotte, Charlotte, NC, United States of America, 4 Wageningen Livestock Research, Wageningen UR, Wageningen, The Netherlands, 5 Visiting Professor Yunnan Agricultural University, Kunming, Yunnan Province, People's Republic of China

☯ These authors contributed equally to this work.
* marinus.tepas@wur.nl (MFWP); junjingj@sina.com, 410457491@qq.com (JJ); yinghuang@ynau.edu.cn (YH)

**Data Availability Statement:** The minimal underlying dataset is contained within the manuscript and its Supporting Information files.

## Abstract

Host defense peptides (HDPs) are an important first line of defense with antimicrobial and immunomodulatory properties. Selection for increased body weight is hypothesized to be related to reduced immune response. We studied the relationships among body weight, age, and the HDP expression patterns in intestine and immune organs. We used chickens with marked differences of body sizes. The non-selected Daweishan mini chickens showed the highest indexes of immune organs and the lowest concentrations of the plasma immune parameters C3, C4, IgA, and IgY, while the commercial Avian broiler showed the opposite results. The Daweishan mini chickens showed the highest mRNA expressions of HDP genes in small intestine followed by the semi-selected Wuding chickens. Compared with local breeds, broiler chickens showed higher mRNA expression of HDP genes in spleen, thymus, and bursa. Body weight and HDP expression levels were negatively correlated in the intestine and positively in the immune organs. Our results indicated that the HDP immune regulatory roles in small intestine acted as first line of defense in innate immunity in local breeds, and as an adaptive immunity in broiler chickens. Selection was associated with different expression expressions of HDP genes in breed-, age-, and organ-specific manners.

Please contact Junjing Jia with additional data inquiries.

**Funding:** CG: National Natural Science Foundation of China (U1702232, 31560618) JJ: National Natural Science Foundation of Yunnan Province (2017IC048 and 2019HC011). The funders had no role in study design, data collection and analysis, decision to publish, or preparation of the manuscript.

**Competing interests:** The authors have declared that no competing interests exist.

## Introduction

As a consequence of prolonged selection for enhanced body weight and growth rate broilers face reduced immune functioning [1–3]. However, it is feasible to breed broilers for enhanced innate immune reactivity [4]. Therefore, a relation between body weight and regulation of immune functions was expected. The host defense *peptides* (HDPs) are proteins that regulate the functioning of the different forms of the immune system. Most HDP genes have the capacity to recruit and activate immune cells, facilitating the resolution of inflammation [5, 6].

The HDP genes are classified in several families and display antimicrobial potential against a broad spectrum of microorganisms, including bacteria, fungi, parasites and viruses [7]. The HDP peptides are small, cationic molecules that act as first line of defense [8, 9]. Many HDP genes are synthesized from the host phagocytic and mucosal epithelial cells that compete with microbes in their environment. Mature HDP peptides are broadly active to control the gram positive and negative bacteria, fungi, viruses, and also some cancerous cells [10]. The HDP genes are expressed in a wide variety of tissues [11–15].

The HDP genes exist in a wide variety of vertebrate species including fish, amphibians, reptiles, birds, and mammals [9, 16–18]. Cathelicidins and β-defensins are two major families of HDP genes in avian species. The first two avian β-defensins (*AvBDs*), also known as gallinacins (*Gals*) 1–2, were isolated in 1994 from chicken heterophils [19–21], an equivalent of mammalian neutrophils. More than a dozen unique *β-defensin* genes are present, and no α- or θ-defensins exist in birds. With the completion of the chicken genome sequencing, a large number of additional chickens β-defensin genes were independently reported [22, 23]. As links between innate and adaptive immunity, many defensins have chemotactic effects on lymphocytes, dendritic cells, and monocytes [24]. In various bird species, *AvBD* genes (or *Gals*) have been found in chickens [20–22, 25–27], turkeys [20, 21], ducks [28, 29], Chinese painted quails [30], zebra finches [31], and mallards [25].

We previously reported that the Daweishan mini chicken breed showed higher indexes of immune organs, especially for index of bursa, and higher expression of genes involved in specific immune traits in muscle tissue than the Wuding chickens [32]. The aim of the present investigation was to investigate the effect of body weight and age on the profiles of expression pattern of HDP genes in different immune organs and their association with innate immune function or adaptive immunity in chickens.

## Materials and methods

### Animal experimentation ethical statement

This study was carried out in strict accordance with the recommendations in the Guide for the Care and Use of Laboratory Animals published by the US National Research Council [33]. The protocol was approved by the Committee on the Ethics of Animal Experiments of the Yunnan Agricultural University Animal Care and Use Committee (approval ID: YNAU#0010).

### Chicken, diet and housing

One-day-old Daweishan mini chickens and Wuding chickens (Local Native breeds in Yunnan Province of P. R. China) were purchased from the Chicken Farm of the Yunnan Agricultural University. Avian broiler chickens of 1-day-old were purchased from the Kunming Zhengda Group (Kunming, Yunnan, P. R. China).

A total of 180 chickens one day of age (60 from each chicken line) were reared under standard conditions on a starter diet as Period I (20.6% CP and 12.8 MJ /kg ME) to 30 d of age, and then 20 chickens from each line were sacrificed at week 4. From 30 d of age on, 40 chickens from each breed were fed a regular diet as Period II (18.4% CP and 12.5 MJ /kg ME) to 60

days (week 8) or 90 days (week 12), and 20 chickens were sacrificed each time point. Diet content was consistent with the formulation to meet NRC 1994 [34] and Chinese Chicken Feeding Standard [35] recommendations. The compositions of diets are shown in Table 1. The chickens had free access to feed and water during the entire rearing period. The temperature was maintained at 35°C for the first 2 days, and then decreased gradually to 22°C (45% humidity) until 30 days and thereafter maintained at 22°C until the end of the experiment at day 90; the light was controlled by fluorescent lighting with a light: dark cycle of 12 h each.

## Slaughter procedure and measurements

The body weight (BW) of the chickens was measured in the morning following a 16 h fasting period at every other week starting at 1 d of age. Feed was withdrawn 16 h and water 12 h before slaughter. The chickens were weighed in a transport box, which was placed on a tared digital scale (Shanghai Yizhan Weighing Apparatus Limited Company, YZ 0.01g-10kg, China). Chickens were slaughtered by cervical dislocation in accordance with the National Experimental Animal Slaughter Standard of China. The Spleen, thymus and bursa of Fabricius were isolated and weighed to calculate the average index of immune organs at each time point. The small intestine was sampled in the mid duodenum.

## Determination of plasma immune parameters

Blood samples were taken from each chicken at all experimental time points via the jugular or wing into vials containing EDTA, and kept on ice. Plasmas were isolated by centrifugation at

**Table 1. Composition of the Period I and Period II (g/kg, air dry) diets used in the experiment.**

| Dietary Component | Period I[1] | Period II[1] |
|---|---|---|
| Maize | 545.0 | 580.0 |
| Soy protein | 190.0 | 167.0 |
| Toasted soybean | 140.0 | 80.0 |
| Fish meal | 35.0 | 20.0 |
| Wheat bran | 30.0 | 100.0 |
| Soya oil | 25.0 | 18.0 |
| $CaHPO_4 \cdot 2H_2O$ | 12.4 | 12.4 |
| Stone meal | 11.7 | 11.5 |
| Lysine | 2.2 | 2.2 |
| Methionine | 1.6 | 1.6 |
| Salt | 2.1 | 2.3 |
| Minerals and vitamins mix[2] | 5.0 | 5.0 |
| Metabolism Energy (MJ/Kg) | 12.8 | 12.6 |
| Crude protein (CP) | 205.5 | 183.5 |
| Crude fat | 54.1 | 56.5 |
| Calcium | 10.5 | 9.8 |
| Available phosphorus | 6.8 | 6.5 |
| Lysine | 13.5 | 12.1 |
| Methionine + Cysteine | 8.8 | 7.8 |

1: Period I is age 1–30 days; Period II is older than 30 days of age; 2: Supplied per kilogram of diet: antioxidant, 100 mg; biotin, 0.3 mg; vitamin A, 12,000 IU; vitamin D3, 3000 IU; vitamin E, 18.75 mg; vitamin K3, 2.65 mg; vitamin C, 12.6 mg; cyanocobalamin, 0.025 mg; folic acid, 2.2 mg; niacin, 35 mg; pyridoxine, 6 mg; riboflavin, 9 mg; thiamine, 3.0 mg; choline chloride, 600 mg; Co, 0.3 mg; Cu, 12 mg; Fe, 50 mg; I, 1 mg; Mn, 125 mg; Mo, 0.5 mg; Se, 200 μg; Zn, 60 mg.

4˚C, and stored at -20˚C prior to enzyme-linked immunosorbent assay (ELISA). Plasma levels for chicken C3, C4, IgA, IgM, and IgY were measured using an ELISA kit (Shanghai Yeyuan Biotechnology Company, China) and with an ELISA Reader (Thermo Fisher, Varioskan Flash, USA) following the manufacturers' instructions. Measurements were done in duplicate. Purified chicken C3, C4, IgA, IgM, and IgY antibodies were used to coat microtiter plate wells. The within-run variations in our laboratory were between 4% and 6%, and between-run variations were 5%.

### Expression analysis of chicken HDP Genes

Samples of small intestine, spleen, thymus, and bursa of Fabricius of chickens were collected and placed in sterile tubes (RNase-free), which were then immediately immersed in liquid nitrogen prior to storage at -80˚C until use. Primers (Table 2) were designed to measure the expression of the candidate and reference genes using the software Primer5 (Premier Biosoft international, Palo Alto, CA, USA). Primer specificity was established using the Basic Local Alignment Search Tool (BLAST) from the National Center for Biotechnology Information (http://www.ncbi.nlm.nih.gov/BLAST/). Primers were commercially synthesized (Shanghai Shenggong Biochemistry Company P. R. China). Tissue homogenization, RNA isolation, and RT-qPCR were performed as well as mRNA expression normalization as previously described [32, 36, 37].

### Correlation analysis

Correlations between the BW of the chickens and the HDP mRNA expression in each of the tissues of the three chicken breeds at different growing periods was analyzed by the Spearman correlation procedure (SAS, CORR procedure, SAS version 9.3, Cary, NC, USA). Spearman correlation was used instead of Pearson correlation because Spearman correlation is a non-parametric statistical method and has a wide range of application, which does not require the distribution of original variables. The correlations between the body weight (BW) and mRNA expression of HDPs in different organs were analyzed, while the body weight (BW) and mRNA expression of HDPs were from three chicken breed and different ages (4, 8 and 12 weeks-old), and the original data were not normal distributed. We analyzed all breeds together to compare the chicken varying from no selection to heavily selected for body weight.

### Statistical analysis

Statistical analyses of blood immune parameters, indexes of immune organs concentrations, and HDP genes mRNA abundance in tissues among breeds at the same age, and among different ages within a breed were performed by ANOVA where appropriate using SPSS 22.0 software package on normalized data in Excel (IBM Corp, Armonk, NY). Data are expressed as the mean ± SEM at each age for each breed. Statistical significance of difference between breeds at an age was labeled by * for $p < 0.05$, and ** for $p < 0.01$. Statistical significance of difference between different ages within a breed is labeled by lower-case letters for $p < 0.05$, and super-case letters for $p < 0.01$.

## Results

All raw PCR and ELISA data can be found in the S1 Data.

### Blood immune parameters and indexes of immune organs

Avian broilers showed the highest serum complement C3 concentrations while Daweishan mini chickens exhibited the lowest concentrations during the 4–12 weeks experimental period

**Table 2. Primer sequences for the target genes and their annealing temperatures.**

| Genes | Gene accession number | Sequence of Primers(5'>3') | Annealing Temperature(°C) |
|---|---|---|---|
| *β-Actin* | NM_205518.1 | F: TGGACTCCTACAACCAACGG | 59.7 |
| | | R: CATCCTCCTTGAACTCGCAG | |
| *AvBD1* | NM_204993.1 | F: TACCTGCTCCTCCCCTTCAT | 59.7 |
| | | R: GATGAGAGTGAGGGAAGGGC | |
| *AvBD2* | NM_001201399.1 | F: CTCTCCTCTTCCTGGCACTC | 59.7 |
| | | R: ATTTGCAGCAGGAACGGAAC | |
| *AvBD3* | NM_204650.2 | F: GTGTACCTGCTCATCCCCTT | 59.7 |
| | | R: GTTGTTTCCAGGAGCGAGAA | |
| *AvBD4* | NM_001001610.2 | F: CATCGTGCTCCTCTTTGTGG | 59.2 |
| | | R: GAATACTTGGGACGGCATAGC | |
| *AvBD5* | NM_001001608.2 | F: CTTTGCTGTCCTCCTCCTGA | 59.2 |
| | | R: AGTCTTCCTTGGAGCAGAGG | |
| *AvBD6* | NM_001001193.1 | F: GCTGCTGTCTGTCCTCTTTG | 59.7 |
| | | R: CCACTGCCACATGATCCAAC | |
| *AvBD7* | NM_001001194.1 | F: TCTGTCCTCTTTGTGGTGCT | 58.0 |
| | | R: CAGCTCCTCCATCCCCTTG | |
| *AvBD8* | NM_001001781.1 | F: GTTCTCCTCACTGTGCTCCA | 56.0 |
| | | R: TTAGTCGTACACAGTCCGGC | |
| *AvBD9* | NM_001001611.2 | F: GCTGTTCTCTTCTTCCTCTTCC | 57.8 |
| | | R: CCCATTTGCAGCATTTCAGC | |
| *AvBD10* | NM_001001609.2 | F: AGATCCTCTGCCTGCTCTTC | 58.0 |
| | | R: TGGCACAGCAGTTTAACAGC | |
| *AvBD11* | NM_001001779.1 | F: CATCCGTTCCAAAGTCTGCC | 59.2 |
| | | R: CTTTGCAGCACGTCCATCTT | |
| *AvBD12* | NM_001001607.2 | F: ATCTTCATCTCCCTGCTCGC | 59.7 |
| | | R: TCAGGTCTTGGTGGGAGTTG | |
| *AvBD13* | NM_001001780.1 | F: GATCCTCCAGCTGCTCTTTG | 59.0 |
| | | R: GTTGGAAAAGGGCTGCTTGG | |
| *AvBD14* | NM_001348511.1 | F: TGGGCATATTCCTCCTGTTTCT | 58.0 |
| | | R: TTGCCAGTCCATTGTAGCAG | |
| *CATH1* | NM_001001605.3 | F:TGGACTCCTACAACCAACGG | 59.7 |
| | | R:CATCCTCCTTGAACTCGCAG | |
| *CATH2* | NM_001024830.2 | F: TGGACTCCTACAACCAACGG | 59.7 |
| | | R:TCTCCTTGAAGTCGCAGTCG | |
| *CATH3* | NM_001311177.1 | F: GCACAACCTCAACTTCACCA | 59.7 |
| | | R: CGTGTTGATGGCCACTGG | |

(Fig 1A). There was no significant difference in serum complement C3 and C4 concentrations between the two local chicken breeds throughout the experimental period (Fig 1A and 1B). In addition, the broilers showed the highest serum C4, IgA, and IgY concentrations while Daweishan mini chickens exhibited the lowest at 4, 8, and 12 weeks (Fig 1B, 1C and 1D). Within each of the chickens, the highest serum complement C3 and C4, IgA and IgM concentrations were found at week 4, and decreased with increasing age.

No significant difference for index of spleen was measured during the whole experimental period among chickens (Fig 1f). However, the Daweishan mini chickens showed the highest indexes of thymus and bursa while the broiler chickens exhibited the lowest throughout experimental period (Fig 1G and 1H).

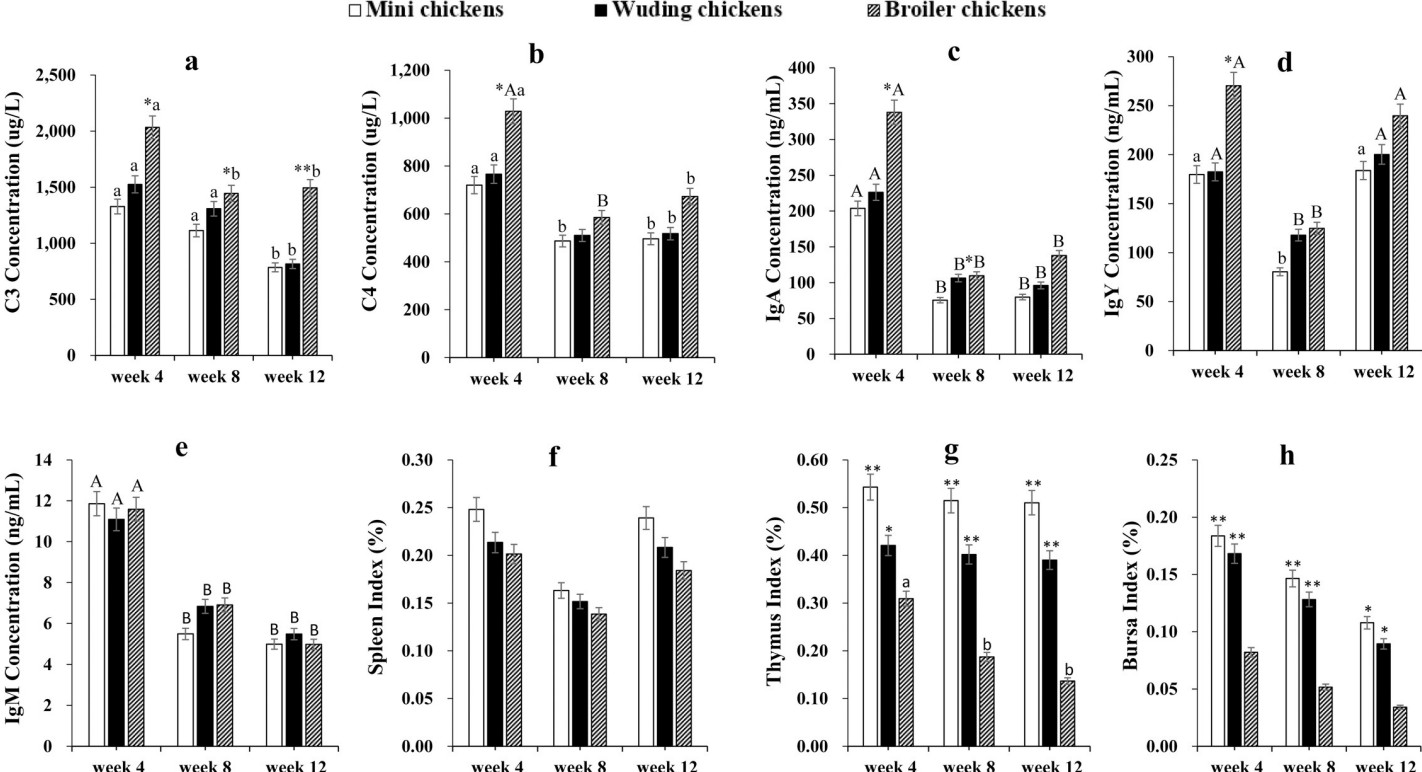

**Fig 1. Blood complements, Igs, and indexes of immune organs.** Comparative analysis of blood complements C3 (a), C4 (b), IgA (c), IgY (d), IgM (e) and indexes of spleen (f), thymus (g), and bursa of Fabricius (h) of three chicken types at weeks 4, 8, 12. Twenty Chickens from each chicken type at three time points were measured. Measurements were done in duplicate. Spleen (f), thymus (g), and bursa of Fabricius (h) were weighed to calculate the average index of immune organs at each time point. Statistical significance of difference between chickens is labeled by * for $p < 0.05$, and ** for $p < 0.01$. Statistical significance of difference between different ages within a chicken is labeled by lower-case letters for $p < 0.05$, and super-case letters for $p < 0.01$.

## HDP mRNA expressions in the small intestine

In the small intestine, mRNA expression of 17 HDP genes were detected with expression level varying with chickens and age. The highest levels of *AvBD 1–4, 7, 11–14* mRNA expression were observed at week 4. All genes exhibited the same expression patterns at week 8 and week 12 exception for *AvBD* 11 and *12*, which showed moderate mRNA expression levels. Moderate levels of *AvBD 6, 8–10* mRNA expression were observed throughout the experimental periods in all chickens. Only low level of *AvBD 5* mRNA expression was observed throughout the experimental periods in the three chickens. It is interesting to observe that the small intestine *AvBD 3, 7, 13, 14* showed high expression levels in throughout experimental periods in three breeds. In contrast, *AvBD 4, 11, and 12* expression levels decreased with increasing age in all chickens.

Fig 2 shows that the Daweishan mini chickens showed the highest mRNA expression of HDP genes while broilers exhibited the lowest expression of HDP genes in the small intestine (Fig 2A–2C), except for the *CATH 3* gene that showed a higher mRNA expression in the broilers than in the Daweishan mini chickens and the Wuding chickens (Fig 2a). No significant differences were measured for *AvBD 2* and *7* at week 4 (Fig 2A), and *AvBD 5, 7, 8* at week 8 (Fig 2B), and *AvBD 4–8,* and *CATH 1* at Week 12 (Fig 2C) among the three chickens.

## HDP mRNA expressions in the spleen

Fig 3 shows the HDP mRNA expression profiles in the spleen, where the Daweishan mini chickens showed the lowest mRNA levels of HDP genes while broilers exhibited the highest

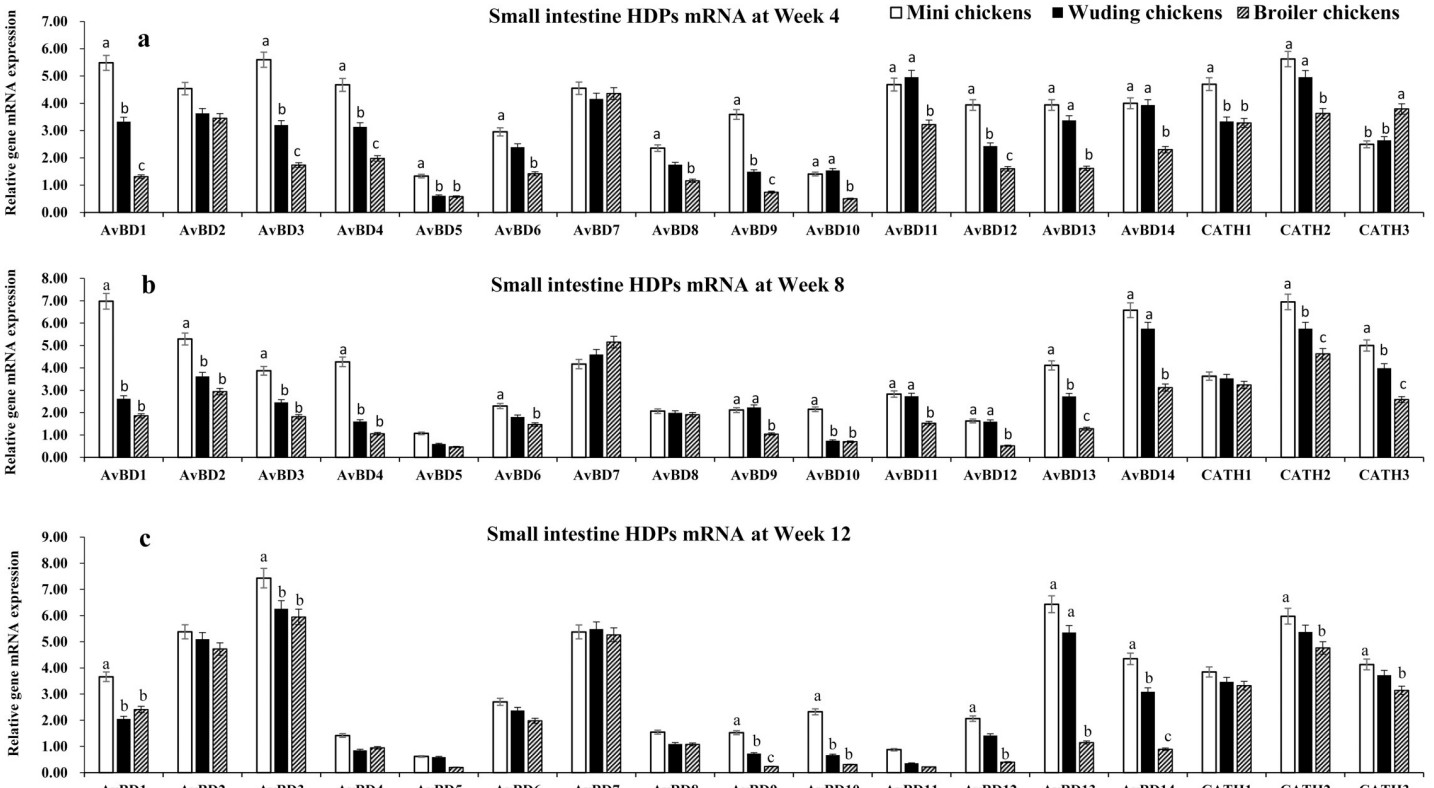

**Fig 2. HDP mRNA expression in the small intestine.** mRNA expression of *AvBD 1–14* and *CATH 1–3* genes in the small intestine of three chickens at weeks 4 (a), 8 (b), and 12 (c). Twenty chickens per type per time point were used and gene expression was investigated with RT-qPCR. The experiment was repeated three times. Data are expressed as the mean ± SEM at each age for each chickens and different letters on top of each bar represent significant differences (p < 0.05) among chickens.

mRNA levels except for the *AvBD 12* and 14 at week 4 (Fig 3a) and *AvBD 12* at week 8 (Fig 3b). Significant differences were measured for *AvBD 1–11*, *13*, and *CATH 1–3* at week 4, for *AvBD 1–11*, *13*, and *CATH 1–3* at week 8 and for all HDP mRNAs at week 12 (Fig 3c). In the spleen mRNA expression of 17 HDP genes were detected with different expression level varying with chickens and age. The highest mRNA levels of *AvBD 1*, *2*, *9*, *13*, and *14* were observed at week 4 and at week 8 for *AvBD 12*, *13* in all three chickens. For the other HDP genes the expression profiles vary with age and breed. The Daweishan mini chickens showed the lowest mRNA levels for all genes except *AvBD 12* in 4 and 8 weeks. The *AvBD 3–7*, *9*, *13*, *14*, and *CATH 1* genes showed high mRNA levels in the Wuding chickens throughout the experimental period and even higher *AvBD* mRNA expressions were observed for broilers. Low to moderate mRNA levels were observed for *AvBD 11*, *12*, and *CATH 2* at weeks 4 and 8, and low mRNA level of *AvBD 11* at week 12.

## HDP mRNA expressions in the thymus

Fig 4 shows that at week 4 the broilers exhibited the highest mRNA levels of HDP genes while the Daweishan mini chickens showed the lowest (Fig 4a) except for the *CATH* 3 gene for which no significant differences were measured among the three chickens. At week 8 the broilers showed the highest mRNA levels of HDP genes while the Daweishan mini chickens showed the lowest HDP mRNA levels except for the *AvBD 4*, *7*, and *14* genes that were higher expressed in the Wuding chickens than in other breeds. No significant differences were measured for the *CATH 1* gene among the three chickens (Fig 4b).

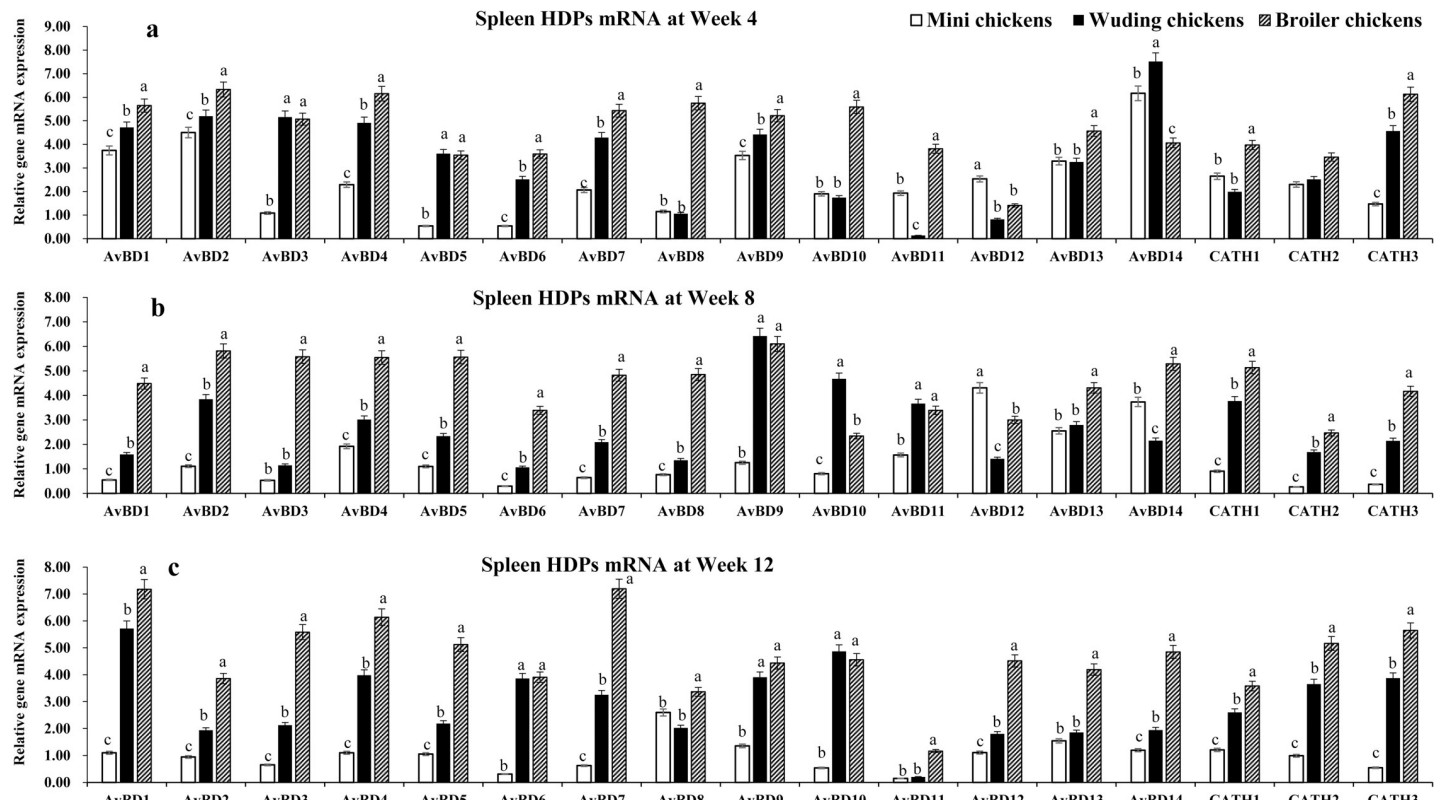

**Fig 3. HDP mRNA expression in the spleen.** mRNA expression of *AvBD 1–14* and *CATH 1–3* genes in the spleen of three chickens at weeks 4 (a), 8 (b), and 12 (c). Twenty chickens per type per time point were used and gene expression was investigated with RT-qPCR. The experiment was repeated three times. Data are expressed as the mean ± SEM at each age for each breed and different letters on top of each bar represent significant differences (p < 0.05) among chickens.

As shown in Fig 4, a total of 17 HDP mRNAs were detected with expression levels varying with breed and age. The highest levels were observed for *AvBD 10* at week 4, for *AvBD 3*, *4*, *13*, and *14* at week 8, for *AvBD 2–5*, *9–11*, and *13* at week 12 and only low mRNA levels of *CATH 3* were observed at week 4 in all three chickens. Overall, the broilers exhibited moderate to high level of HDP genes mRNA expression except for *CATH 3* at week 4, *CATH 1–3* at weeks 8 and 12. In the thymus of the Daweishan mini chicken an age dependent decreased expression was observed in *AvBD genes* except for *AvBD 12* and *14*.

## HDP mRNA expressions in the bursa of Fabricius

Spleen and bursa showed the similar expression profiles for most HDP genes (Fig 5). The broilers exhibited the highest HDP mRNA levels while Daweishan mini chickens showed the lowest HDP mRNA levels except for *CATH 2*, and *3* at weeks 4 and 8 (Fig 5b) and for *CATH 3* at week 12 (Fig 5c) where a significant higher expression in the Daweishan mini chickens than other two chickens was found.

A total of 17 HDP mRNAs were detected with expression levels varying with chickens and age. Overall, broilers exhibited moderate to high level of HDP genes mRNA expression throughout the experimental periods. In the two local breeds moderate to high levels of HDP genes mRNA expression were observed only for *AvBD 6*, *12–14*, *CATH 1–3* at week 4; for *AvBD 2–4 and 13*, *CATH 1–3* at week 4 week 8, for *AvBD 2–5*, *8–10*, *12*, *13*, and *CATH 1–3* at week 12. Only low levels of *AvBD 13* mRNA expression was observed throughout the experimental period in all three chickens.

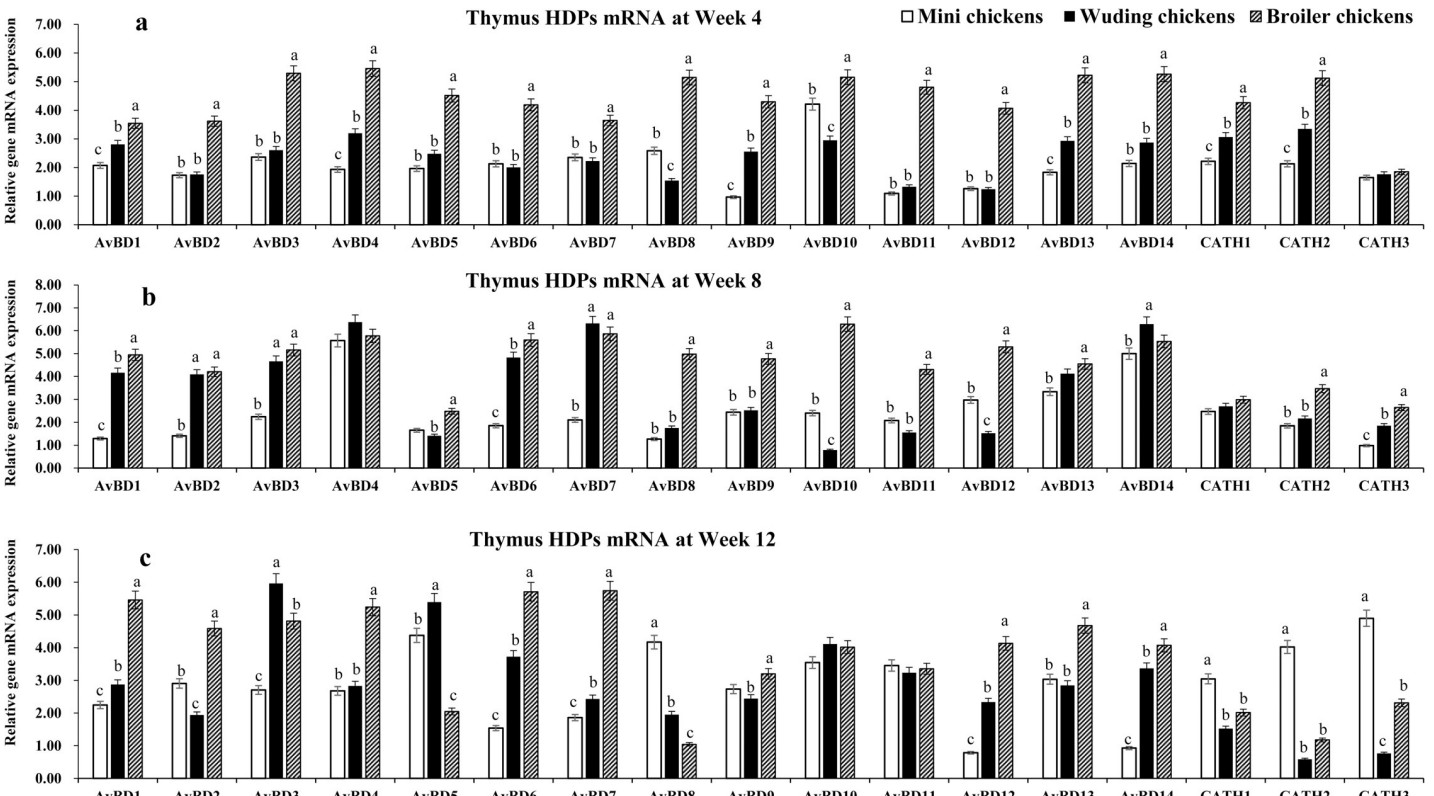

**Fig 4. HDP mRNA expression in the thymus.** Comparison analysis of mRNA expression of *AvBD 1–14* and *CATH 1–3* genes in the thymus of three chickens at weeks 4 (a), 8 (b), and 12 (c). Twenty chickens per type per time point were used and gene expression was investigated with RT-qPCR. The experiment was repeated three times. Data are expressed as the mean ± SEM at each age for each breed and different letters on top of each bar represent significant differences (p < 0.05) among chickens.

## Correlations between chicken body weight and HDP mRNA expression

Table 3 shows the correlations between the chicken BW and the HDP mRNA expression in four organs. The majority of the correlations for the small intestine were negative, while the majority of the correlations for the immune organs were positive. The *AvBD5* HDP was significantly correlated in three out of four organs, while the mRNA expressions of *AvBD4*, *8*, and *CATH1* were not correlated in any of the organs.

Correlations between BW and the mRNA expression of HDP genes (total 17 HDP genes) in the small intestine, thymus, bursa and spleen of three chicken breeds at different growing period were analyzed using the spearman correlation procedure with the CORR procedure of SAS (SAS version 9.3). The correlations were considered significant if P< 0.05.

## Discussion

### The tissue-specific expression of HDP mRNAs

In the chicken genome, the *AvBD* genes have been clustered on chromosome 3 [10, 31, 38]. The expression of *AvBD* genes *1–7* have been reported in bone marrow, digestive tract, and respiratory epithelium, while the *AvBD* genes *8–14* have been reported in liver and the gastro-urinary system [22, 25]. In the present study we identified different patterns of the *AvBD 1–14*, and 3 cathelicidin genes (*CATH 1–3*) in the small intestine, and three immune-organs: spleen, thymus, and bursa, of chickens related to BW and age. Interestingly, the correlations between

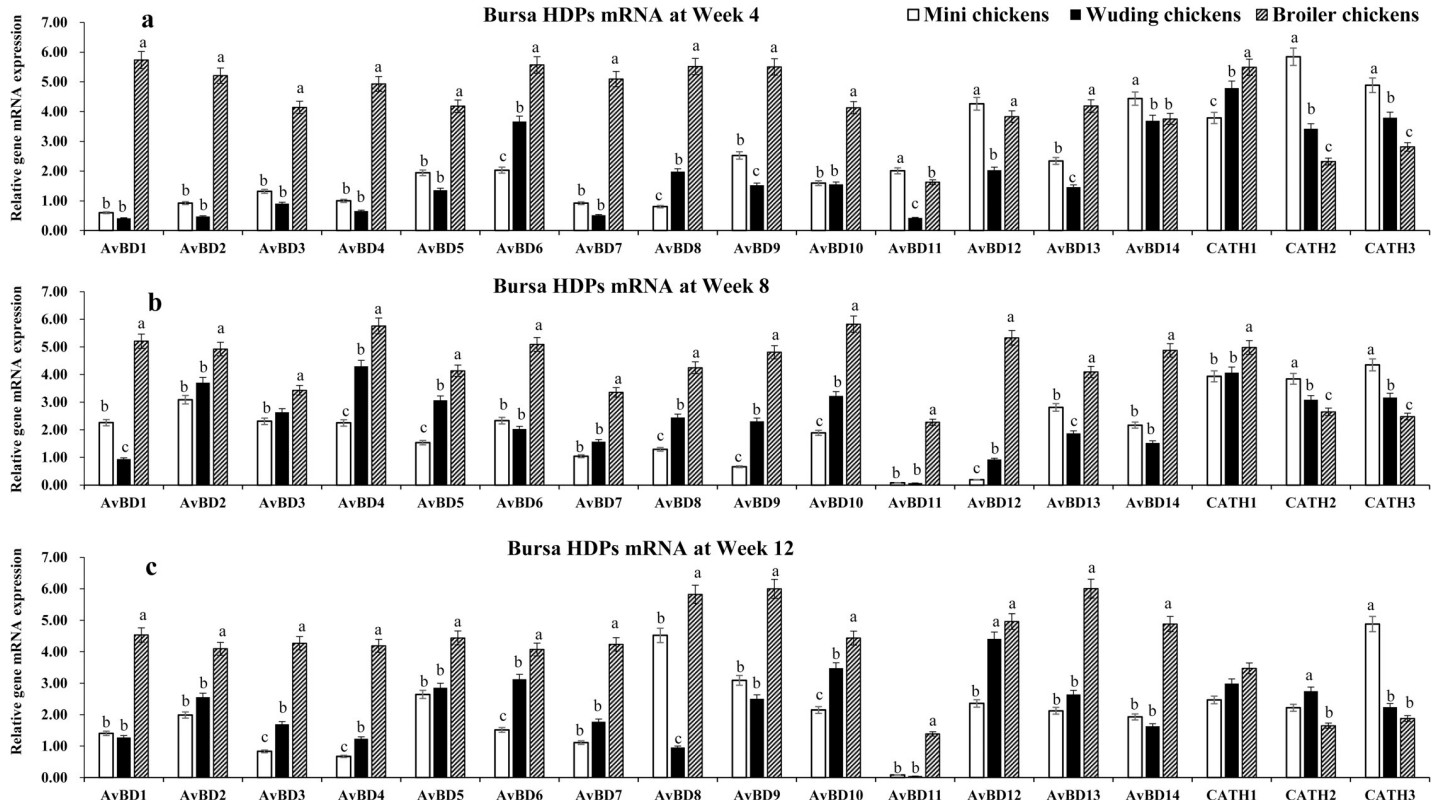

**Fig 5. HDP mRNA expression in the bursa of Fabricius.** Comparison analysis of mRNA expression of *AvBD 1–14* and *CATH 1–3* genes in the bursa of Fabricius of three chickens at weeks 4 (a), 8 (b), and 12 (c). Twenty chickens per type per time point were used and gene expression was investigated with RT-qPCR. The experiment was repeated three times. Data are expressed as the mean ± SEM at each age for each chicken and different letters on top of each bar represent significant differences (p < 0.05) among chickens.

BW and the expression levels of the HDP mRNAs differs between the small intestine and the three immune organs. The correlation shows that increased BW is accompanied by decreased HDP expression in the small intestine, but with increased HDP expression in the immune organs. The difference may relate to the main function of the intestine, i.e. uptake of nutrients from the food. Faster growing animals reaching a higher BW need to develop their intestine in advance of the growing period. This may come with a reduced immune development, for example due to energy partitioning towards the nutrient uptake function. This also makes the intestine more vulnerable to infection. The more important because this organ is in close contact with pathogens from outside the body. This may be compensated by improved immune function from the three immune organs.

## HDP genes in the small intestine

The gastrointestinal tract is constantly exposed to a wide range of potential pathogens and especially susceptible to infection. The innate immune response is the first line of defense against infection. The epithelial cells of the intestinal mucosa facilitate efficient host defense by producing the HDP genes [39], which plays a significant role in the chicken innate immune system, providing the first line of defense against potential pathogens [21, 27, 40, 41]. We found that the Daweishan mini chickens showed the highest mRNA levels of HDP genes mRNA in small intestine tissue while Avian broilers exhibited the lowest mRNA levels of HDP

**Table 3. Correlations between the body weight (BW) and mRNA expression of HDP genes in different organs.**

| HDP genes | BW Small Intestine | BW Thymus | BW Bursa | BW Spleen |
|---|---|---|---|---|
| *AvBD1* | -0.511 | 0.844** | 0.64 | 0.619 |
| *AvBD2* | -0.145 | 0.718* | 0.576 | 0.195 |
| *AvBD3* | 0.038 | 0.602 | 0.688* | 0.616 |
| *AvBD4* | -0.664 | 0.388 | 0.592 | 0.636 |
| *AvBD5* | -0.730* | -0.049 | 0.781* | 0.787* |
| *AvBD6* | -0.479 | 0.815** | 0.511 | 0.682* |
| *AvBD7* | 0.600 | 0.676* | 0.657 | 0.738* |
| *AvBD8* | -0.499 | -0.005 | 0.597 | 0.515 |
| *AvBD9* | 0.682* | 0.535 | 0.744* | 0.375 |
| *AvBD10* | -0.652 | 0.435 | 0.803** | 0.395 |
| *AvBD11* | -0.686* | 0.521 | 0.403 | 0.095 |
| *AvBD12* | -0.791* | 0.719* | 0.649 | 0.555 |
| *AvBD13* | -0.584 | 0.578 | 0.853** | 0.474 |
| *AvBD14* | -0.708* | 0.281 | 0.500 | 0.044 |
| *CATH1* | -0.490 | -0.261 | -0.021 | 0.593 |
| *CATH2* | -0.406 | -0.280 | -0.613 | 0.696* |
| *CATH3* | -0.285 | 0.102 | -0.788* | 0.560 |

*: Significance of correlation at P<0.05.

**: Significance of correlation at P<0.01.

genes. A higher expression may relate to better immunity and may relate to either selection response directly, or via selection-induced nutrient repartitioning affecting immune responses via the gene levels. Especially in the non-selected small sized Daweishan mini chickens strong innate immunity associated with specific tissue expressions of HDP genes in small intestine. Tissue-specific expression, immune regulation and corresponding biological functions in the innate immune systems has been reported before [22, 25, 26, 42].

The expression of *AvBD 1, 2, 4, 6, 7, 9,* and *13* genes in the small intestine has been reported [27, 43, 44]. The *AvBD 1, 2, 4, 5, 7,* and *10* genes were moderately expressed at 3 weeks of age [22], *AvBD 13* exhibited low expression level at 8 week [26].

We found high levels of *AvBD 1–3* mRNA expression throughout the experimental period together with an age-related increase for small intestine *AvBD 4* in the three chickens. The latter has been observed before [44]. Contrasting, *AvBD 1* and *AvBD 2* mRNA levels were reported to decrease during the first week and increase during the second week post-hatch [43]. We found that the *AvBD 7, 13,* and *14* showed high mRNA expression at weeks 4, 8, and 12 in the small intestine. This confirms previous reports for *AvBD 13* [27].

Moderate levels of *AvBD* genes *6, 8–10* mRNA expression were observed throughout the experimental periods in all chickens. Only low mRNA level of the *AvBD 5* was observed throughout the experimental periods in the three breeds, while the *AvBD 5* gene mRNA was not detected in the small intestine tissue in previous reports [27, 43, 44]. This may relate to the methods used, since the PCR method is a very sensitive method.

*CATH 1–3* exhibited an age-dependent expression profile peaking at 4 weeks of age, similar to previous reports in other chickens [42]. We showed that *AvBD 2* and *7* exhibited a moderate expression level but that the *AvBD 1, 4, 5,* and *10* genes exhibited low or no expression at 4 weeks in broiler. The *AvBD 13* gene showed a similar expression pattern [26].

Concluding, it may be possible that the use of unselected unique chickens and commercial broilers with a long selection background in our study created a contrast that highlights processes not measurable in studies with commercial broiler lines only. This may show that the biological regulatory mechanism can be affected by experimental conditions including selection background, and age of chickens.

## HDP genes immune organs

Tissue-specific expression patterns for *AvBD* mRNAs in immune organs, the digestive tract and the urogenital tract have been reported [42, 45, 46]. We found that the HDP mRNA expression in general showed the profile bursa > spleen > thymus. The localization of the organs may be involved here [45]. The bursa is located near the cloaca and therefore continuously exposed to microorganisms. The local moderate to high expression of several *AvBD* genes in this organ may regulate its functioning [42, 45, 46]. Different levels of Avian HDP genes mRNA expression have been reported in the digestive tract, urogenital tract, and immune organs with organ-specific expression patterns [22, 25, 26]. Only *AvBD 9* and *11* genes mRNA were detected in the thymus with weak expression, the *AvBD 2*, *8* weak expressions and *AvBD 13* strong expression in spleen, low-high level *AvBD 1–10*, *12*, and *13* mRNA expression in bursa [27, 45, 47].

We found 14 *AvBD* genes and 3 *CATH*s expressed in small intestine, spleen, thymus, and bursa with breed, age, and organ-specific expression patterns. We found that compared to the Daweishan mini chickens the Avian broiler chickens showed high level expression pattern of HDP genes mRNA in immune organs. The function of HDP genes may related more to adaptive immunity in broiler chickens, which agrees with previous reports that the HDP genes have been proposed to bridge the innate and adaptive immune responses by inducing the differentiation of dendritic cells and macrophages [48].

We found that the Avian broilers showed the highest serum complement C3, C4, IgA, and IgY concentrations while Daweishan mini chickens exhibited the lowest levels for these parameters. Immunoglobulins (Igs) are essential molecules for the animal adaptive immune response and are expressed only in jawed vertebrates, including fish, amphibians, reptiles, birds, and mammals [49]. Three immunoglobulin classes (IgA, IgM and IgY) have been shown to exist in the chicken. They have different roles in the adaptive immune responses and are commonly produced after bacterial or viral infection [50]. HDP genes can act directly on B and T cells which is crucial for influencing subsequent adaptive immune responses. The murine homolog of human *LL37*, *CRAMP* (cathelin-related antimicrobial peptide), was shown to modulate immunoglobulin IgY production in B cells [51]. Thus the HDP genes can act as a link between innate and adaptive immune systems [17, 52].

## Conclusions

We conclude that selection for growth performance and body size has altered the expression profiles of HDP *genes* in a breed-, age-, and organ-specific manner, and that regulation of the regulatory mechanisms of these genes might play an important role in the innate immune systems of chicken breeds.

## Supporting information

**S1 Data. Raw PCR and ELISA data.**
(XLSX)

## Author Contributions

**Conceptualization:** Zhengchang Su, Changrong Ge, Junjing Jia.

**Data curation:** Tengfei Dou, Marinus F. W. te Pas, Junjing Jia.

**Formal analysis:** Alsoufi Mohammed Abdulwahid, Tengfei Dou, Marinus F. W. te Pas, Junjing Jia.

**Funding acquisition:** Changrong Ge, Junjing Jia.

**Investigation:** Zhengtian Li, Irfan Ahmed, Shuai Sun, Tao Li, Lixian Liu, Hua Rong, Kun Wang, Alsoufi Mohammed Abdulwahid.

**Methodology:** Zhengtian Li, Irfan Ahmed, Zhiqiang Xu, Shuai Sun, Tao Li, Lixian Liu, Hua Rong.

**Project administration:** Changrong Ge, Junjing Jia.

**Resources:** Xia Zhang, Shixiong Yan, Wenyuan Hu, Ziqing Jiang.

**Supervision:** Changrong Ge, Marinus F. W. te Pas.

**Validation:** Dahai Gu, Yong Liu, Xiaohua Duan, Qihua Li, Shanrong Wang, Zhengchang Su, Ying Huang.

**Visualization:** Tengfei Dou, Marinus F. W. te Pas, Junjing Jia.

**Writing – original draft:** Zhengtian Li, Tengfei Dou, Marinus F. W. te Pas, Junjing Jia.

**Writing – review & editing:** Marinus F. W. te Pas, Junjing Jia.

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
