## [Decision Letter · Decision Letter 0]

8 Oct 2020

PONE-D-20-26122

Profiles of Expression pattern and tissue distribution of host defense peptides genes in different chicken (Gallus gallus) breeds related to body weight

PLOS ONE

Dear Dr. te Pas,

Thank you for submitting your manuscript to PLOS ONE. After careful consideration, we feel that it has merit but does not fully meet PLOS ONE’s publication criteria as it currently stands. Therefore, we invite you to submit a revised version of the manuscript that addresses the points raised during the review process.

Please pay particular attention to the questions asked by Reviewer 1 concerning: (1) the normalizing the qRT-PCR data and (2) the your statistical analysis of all of your data.  It is obvious that the review has serious doubts about the data and data analysis. Your responses must be detailed and provide the required details to satisfy the reviewer the validity of the data and the statistical analysis of the data.

We look forward to receiving your revised manuscript.

Kind regards,

Michael H. Kogut, Ph.D.

Academic Editor

PLOS ONE

Journal Requirements:

Reviewers' comments:

Reviewer's Responses to Questions

**Comments to the Author**

1. Is the manuscript technically sound, and do the data support the conclusions?

Reviewer #1: Yes

Reviewer #2: Yes

2. Has the statistical analysis been performed appropriately and rigorously? 

Reviewer #1: No

Reviewer #2: I Don't Know

3. Have the authors made all data underlying the findings in their manuscript fully available?

Reviewer #1: Yes

Reviewer #2: Yes

4. Is the manuscript presented in an intelligible fashion and written in standard English?

Reviewer #1: No

Reviewer #2: No

5. Review Comments to the Author

Reviewer #1: The manuscript by Li, Z., et al. compared the plasma concentrations of several immune parameters (C3, C4, IgA, and IgY) as well as the expression levels of 17 chicken HDP genes in the small intestine, spleen, thymus, and bursa of three different breeds of chickens of varying maturing sizes. A few methodologies are a little confusing and some of the conclusions are not well supported by the data. The following are my specific comments:

1. Although the authors rightfully indicated that HDP gene expressions are highly tissue-specific, each of 14 chicken beta-defensins and 3 cathelicidins was surprisingly found to be expressed in all four tissues examined including the small intestine, spleen, thymus, and bursa. Please provide the literature support or experimental evidence to show all 17 chicken HDP genes are expressed in each of the four tissue types.

2. The expression levels of chicken HDP genes was examined in the small intestine. What is the specific segment of the small intestine (duodenum, jejunum or ileum)? For each segment, what is the specific section (proximal, middle, or distal)? These details need to be provided.

3. In qPCR experiments, a total of 720 samples (= 4 x 60 x 3) were collected from 4 tissue types in each of 60 animals (in 3 breeds) at 3 different ages (weeks 4, 8, and 12). For each sample, qPCR was performed with 17 chicken HDP genes and one reference gene. Each qPCR reaction was said to repeat 3 times based on figure legends. So in the end, a total of 720 x 18 x 3 = 38,880 qPCR reactions were run. Please describe in detail how total RNA were isolated from 720 samples and how 38,880 RT-qPCR reactions were set up, and how data were normalized across hundreds of different PCR plates in order to ensure that the differences in HDP expression levels are not due to technical variations?

4. Spearman correlation analysis was used to study the correlation between body weight and the expression levels of each of the 17 chicken HDP genes in each of the small intestine, spleen, thymus, and bursa. Please justify why Spearman correlation, not Pearson correlation, was performed. Since HDP expression patterns in three breeds of chickens are significantly different from each other, what is the reason to combine all three breeds? Does analyzing each breed make more sense? Many of the correlation coefficients are nearly 0.65, but surprisingly still insignificant. Please double-check your statistics, particularly given your large sample size (n = 60).

5. "Host defense peptides" or "HDP" doesn’t need to be italicized in the text because it is not the name of a specific gene. Only a symbol (or abbreviation), not full name, for a gene is italicized. Therefore, "avian beta-defensin 1" or "cathelicidin 2" don’t need to be italicized only when you write AvBD1 or Cath2. Please make corrections in the text.

6. All figures' resolutions are low and statistics are barely discernable.

7. Move all tables and figure legends to the end of the text.

8. Grammatical errors are scattered through the text and need to be corrected.

9. Lines 24-25 - "Selection for increased body weight is suggested to be related to reduced immune response". Please provide literature support.

10. Lines 32-34 – "These results indicated that the HDP immune regulatory roles in small intestine acted as first line of defense in innate immunity in local breeds, and as an adaptive immunity in broiler chickens". I don’t understand the statement. Please explain how it is supported by the experimental results?

11. Line 35 – "Selection was associated with the expression profiles of HDP genes in a chicken, age, and organ-specific manner". Change it to "Selection was associated with different expression expressions of HDP genes in breed-, age-, and organ-specific manners".

Reviewer #2: Lines 45 – 57: It is important to differentiate between HDP genes and the peptides for which they encode. As these lines seem to be referring to the molecules rather than the genes, the phrase “HDP genes” should be replaced with “HDPs” or “HDP peptides”

Lines 78 – 79: “20 chickens from each chickens” is confusing. Should be changed to “20 chickens from each line”

Line 105: Reads “jugular of wing”. I believe it should be changed to “jugular or wing”

Lines 134 – 135: To my knowledge, repeated t-test is never an appropriate analysis. ANOVA should be used in all situations that compare more than 2 groups. If you keep the repeated t-test please clarify in which situations it was used.

Lines 145 – 147: Seems to be some discrepancy between text and the figure labels regarding which time points are significant between breeds. For example, text says “There was no significant difference in the serum complement C4 and IgA concentrations at 8 – 12 weeks” but figure indicates IgA significance at week 8. Please double check figures and clarify.

Line 196: Wording “mRNA expression” is a repeat. Please remove.

Line 196 – 197: This seems to be a repeat of lines 188 to 190. In general, the two paragraphs in this section would make more sense if combined into one.

Line 210: Change “exception” to “except”

Line 209 – 214: Please clarify to which week each sentence is referring rather than just including the panels in parenthesis.

Line 219 – 220: This sentence does not make sense. Please revise.

Overall resolution and quality of the figures needs to be improved. They are all fuzzy making the wording and significance indicators difficult to read. Also, please indicate the statistical test used in each situation in the figure legends.

6. PLOS authors have the option to publish the peer review history of their article (what does this mean?). If published, this will include your full peer review and any attached files.

Reviewer #1: No

Reviewer #2: No

---

## [Author Response · Author response to Decision Letter 0]

18 Nov 2020

All responses to the comments of the reviewers - including the changes made in the manuscript - can be found in the rebuttal letter

---

## [Editor Report · Decision Letter 1]

23 Nov 2020

Profiles of expression pattern and tissue distribution of host defense peptides genes in different chicken (Gallus gallus) breeds related to body weight

PONE-D-20-26122R1

Dear Dr. te Pas,

We’re pleased to inform you that your manuscript has been judged scientifically suitable for publication and will be formally accepted for publication once it meets all outstanding technical requirements.

Kind regards,

Michael H. Kogut, Ph.D.

Academic Editor

PLOS ONE
---

## [Editor Report · Acceptance letter]

14 Dec 2020

PONE-D-20-26122R1 

Profiles of expression pattern and tissue distribution of host defense peptides genes in different chicken (*Gallus gallus*) breeds related to body weight 

Dear Dr. te Pas:

I'm pleased to inform you that your manuscript has been deemed suitable for publication in PLOS ONE. Congratulations! Your manuscript is now with our production department. 

Kind regards, 

on behalf of

Dr. Michael H. Kogut 

Academic Editor

PLOS ONE